# Programmed Cell Death-1: Programmed Cell Death-Ligand 1 Interaction Protects Human Cardiomyocytes Against T-Cell Mediated Inflammation and Apoptosis Response In Vitro

**DOI:** 10.3390/ijms21072399

**Published:** 2020-03-31

**Authors:** Woan Ting Tay, Yi-Hsien Fang, Suet Theng Beh, Yen-Wen Liu, Ling-Wei Hsu, Chia-Jui Yen, Ping-Yen Liu

**Affiliations:** 1Institute of Clinical Medicine, College of Medicine, National Cheng Kung University, 1 University Road, Tainan City 70403, Taiwan; woanting1006@gmail.com (W.T.T.); eddiefang0023@gmail.com (Y.-H.F.); 2Institute of Basic Medical Sciences, College of Medicine, National Cheng Kung University, 1 University Road, Tainan City 70403, Taiwan; suettheng@hotmail.com (S.T.B.); greenorange77jh@gmail.com (L.-W.H.); 3Division of Cardiology, Department of Internal Medicine, National Cheng Kung University Hospital, College of Medicine, National Cheng Kung University, 138 Sheng-Li Rd. North District, Tainan 70403, Taiwan; wen036030@gmail.com; 4Center of Cell therapy, National Cheng Kung University Hospital, College of Medicine, National Cheng Kung University, 1 University Road, Tainan 70403, Taiwan; 5Division of Hematology and Oncology, Department of Internal Medicine, National Cheng Kung University Hospital, College of Medicine, National Cheng Kung University, 138 Sheng-Li Rd. North District, Tainan 70403, Taiwan; yencj@mail.ncku.edu.tw

**Keywords:** immune-related cardiotoxicity, PD-1, PD-L1, human embryonic stem cell-derived cardiomyocytes, nivolumab, T-lymphocytes

## Abstract

Aim: Immunological checkpoint therapy is considered a powerful method for cancer therapy and acts by re-activating autologous T cells to kill the cancer cell. Myocarditis cases have been reported in cancer patients after immunological therapy; for example, nivolumab treatment is a monoclonal antibody that blocks programmed cell death-1/programmed cell death ligand-1 ligand interaction. This project provided insight into the inflammatory response as a benchmark to investigate the potential cardiotoxic effect of T cell response to the programmed cell death-1 (PD-1)/programmed cell death ligand-1 (PD-L1) axis in regulating cardiomyocyte injury in vitro. Methods and Results: We investigated cardiomyopathy resulted from the PD-1/PD-L1 axis blockade using the anti-PD-1 antibody in Rockefeller University embryonic stem cells-derived cardiomyocytes (RUES2-CMs) and a melanoma tumor-bearing murine model. We found that nivolumab alone did not induce inflammatory-related proteins, including PD-L1 expression, and did not induce apoptosis, which was contrary to doxorubicin, a cardiotoxic chemotherapy drug. However, nivolumab was able to exacerbate the immune response by increasing cytokine and inflammatory gene expression in RUES2-CMs when co-cultured with CD4^+^ T lymphocytes and induced apoptosis. This effect was not observed when RUES2-CMs were co-cultured with CD8^+^ T lymphocytes. The in vivo model showed that the heart function of tumor-bearing mice was decreased after treatment with anti-PD-1 antibody and demonstrated a dilated left ventricle histological examination. The dilated left ventricle was associated with an infiltration of CD4^+^ and CD8^+^ T lymphocytes into the myocardium. PD-L1 and inflammatory-associated gene expression were significantly increased in anti-PD-1-treated tumor-bearing mice. Cleaved caspase-3 and mouse plasma cardiac troponin I expressions were increased significantly. Conclusion: PD-L1 expression on cardiomyocytes suppressed T-cell function. Blockade of PD-1 by nivolumab enhanced cardiomyocyte inflammation and apoptosis through the enhancement of T-cell response towards cardiomyocytes.

## 1. Introduction

In recent years, immune checkpoint inhibitors (ICIs) have revolutionized cancer treatment due to promising results in advanced tumor cases with traditionally poor prognosis [1]. Immune checkpoints serve as co-inhibitory signals to down-regulate immune response and to promote T cell apoptosis. Cytotoxic T-lymphocytes-associated antigen-4 (CTLA-4), programmed cell death-1 (PD-1), and programmed cell death ligand-1 (PD-L1) are well-known immune checkpoint modulators and are implemented for cancer treatment [2,3].

Paradoxically, enhanced immune response via blocking immune checkpoints can be a double-edged sword. This is due to the over-activated immune system resulting in a spectrum of adverse events. This includes immune-related cardiotoxicity from the off-target effects of ICIs towards the cardiovascular system [4]. Although ICI-related cardiotoxicity is considered a rare case, this can be life-threatening [5]. Two cases of fulminant myocarditis with massive CD4^+^ and CD8^+^ T-lymphocytes infiltration have been reported for patients who received combinational therapy of ipilimumab (anti-CTLA-4 antibody) and nivolumab (anti-PD-1 antibody) [6]. Patients developed cardiogenic shock, and post-mortem analysis displayed CD3^+^ lymphocyte infiltration into the myocardium after nivolumab monotherapy [7].

PD-1 is expressed on T-lymphocytes, B cells, and macrophages upon activation. PD-1 transmits immune suppressive signals by interacting with its ligands—PD-L1 and PD-L2 [8]. PD-L1 is expressed on a wide range of non-hematopoietic cells, including murine and human heart [6,8,9], and thus the PD-1/PD-L1 regulatory system is crucial in maintaining peripheral tolerance [10]. PD-1/PD-L1 plays a pivotal role in regulating cytotoxic T-lymphocyte and effector T-helper cell activities to prevent autoimmune myocarditis [11]. Cardiotoxicity can be happening when PD-1/PD-L1 is interrupted. Nishimura et al. demonstrated PD-1 deficient mice developed spontaneous autoimmune dilated cardiomyopathy [12]. On the other hand, the deletion of PD-1 in Murphy Roths Large (MRL) mice led to lethal autoimmune myocarditis with massive CD4^+^ and CD8^+^ T-lymphocytes infiltration [13]. 

We hypothesized the role of PD-1 and PD-L1 in preventing T-lymphocyte-mediated cardiomyopathy in vitro and in vivo. Since none of the currently available studies have stated the direct effect of anti-PD-1 treatment on cardiomyocytes, we first demonstrated the effect of nivolumab (OPDIVO^®®^), an anti-human-PD-1 monoclonal antibody, on human embryonic stem cell-derived cardiomyocytes (hESC-CMs) without involving immune cells. Using a direct contact co-culture model of hESC-CM with T-lymphocytes, we revealed a PD-1/PD-L1 axis regulating CD4^+^ and CD8^+^ T cell response and the potential of nivolumab to increase T-cell responses against cardiomyocytes.

In the in vivo model, the heart function of tumor-bearing mice was decreased after treatment with an anti-PD-1 antibody. Cardiac histology of anti-PD-1-treated tumor-bearing mice demonstrated a dilated left ventricle, which was resulted by the infiltration of CD4+ and CD8+ T lymphocytes into the myocardium. The expression of PD-L1, interferon-gamma, and tumor necrosis factor-alpha protein was increased significantly in anti-PD-1-treated tumor-bearing mice. Cleaved caspase-3 expression and mouse plasma cardiac troponin I were also raised significantly in the anti-PD-1-treated tumor group. This study provided an in vitro and in vivo platform to study the role of PD-1/PD-L1 axis and how anti-PD-1 inhibition increases T lymphocytes-mediated cardiomyopathy. Our data provided preclinical evidence of the immune-related cardiomyopathy with the administration of anti-PD-1 ICIs, and thus the cardiac condition should be concerned and monitored in cancer patients treated with anti-PD-1 ICIs therapy.

## 2. Methods

### 2.1. Human Embryonic Stem Cell Culture

The Rockefeller University embryonic stem cell line-2 (RUES2) was a kind gift from Dr. Yen-Wen, Liu [Department of Internal Medicine, National Cheng Kung University (NCKU) Hospital]. Before culturing the cells, growth factor-reduced Matrigel (#354230; Corning, Carlsbad, Leicestershire, UK) was added to the culture plate as an extracellular matrix coating. Cells were then seeded on Matrigel-coated plates supplemented with essential-8 (E8) medium (#A1517001; Life Technologies, Carlsbad, CA, USA) and 5 µM Y27632 (ROCK inhibitor; R&D Systems, Minneapolis, Minnesota, USA). E8 medium renewal was performed daily without adding Y27632. RUES2 was passaged every 2–3 days using Accutase (Innovative Cell Technologies, San Diego, CA, USA) for cell detachment.

### 2.2. Cardiomyocytes Differentiation

Cardiomyocytes differentiation was performed as previously described [14]. Briefly, RUES2 cells were expanded to obtain a monolayer on the plate. The medium was replaced by 1 µM CHIR99021 (GSK-3 inhibitor; Tocris Bioscience, Bristol, UK) and supplemented with E8 medium on the next day. This time point corresponded to day 1 of differentiation. On day 0, the medium was exchanged with 100 ng/mL activin A (R&D Systems) in Roswell Park Memorial Institute (RPMI) 1640 medium supplemented with 2% of B-27 minus insulin (Invitrogen, Carlsbad, CA). After 18 h of incubation, the medium was replaced with RPMI/B-27 minus insulin supplemented with 5 ng/mL bone morphogenic protein-4 and 1 µM CHIR99021. On day 3, the medium was replaced with RPMI/B-27 minus insulin supplemented with 5 µM XAV939 (Wnt/β-catenin inhibitor; Tocris Bioscience, Bristol, Bristol, UK). The medium was exchanged with RPMI/B-27 minus insulin on day 5. The medium was replaced with RPMI supplemented with 2% B-27 with insulin (Invitrogen, Carlsbad, CA, USA) on day 7. On day 9, the selection was done by replacing the medium with RPMI 1640 medium without glucose supplemented with 2% B-27 with insulin. Differentiated RUES2-cardiomyocytes (RUES2-CMs) were then detached from the plate by using trypsin-ethylene diamine tetra-acetic acid (Gibco, Thermo Fisher Scientific, Waltham, MA, USA) and re-plated on suitable plates.

### 2.3. Immunocytochemistry

RUES2-CMs were first seeded on the ibidi µ-Slide 8-well chambered slides (Martinsried, Germany). The cells were fixed with 4% paraformaldehyde prior to being blocked with 5% bovine serum albumin (BSA). The cells were stained with α-actinin (1:200; Abcam, Cambridge, UK), cardiac troponin T (cTnT) (1:200; Abcam), and PD-L1 (1:100; Proteintech Group Inc., Chicago, IL, USA) antibodies and incubated overnight at 4 °C. The cells were incubated with appropriate secondary antibodies (Alexa-488 and Alexa-594; 1:500; Invitrogen). For CD4^+^ and CD8^+^ T-lymphocytes, cells were added to cytospin slide chambers with blotters and glass slides attached. The cells were then centrifuged on a glass slide using a cytocentrifuge. The staining process was done by adding the PD-1 antibody (1:100; Proteintech Group Inc., Rosemont, IL, USA) and then appropriate secondary antibodies (Alexa-488 and Alexa-594; 1:500; Invitrogen). 4’,6-diamidino-2-phenylindole (DAPI) was utilized to observe cells’ nuclei.

### 2.4. MTT Assay

A total number of 1 × 10^4^ RUES2-CMs were seeded on 96-well Matrigel-coated-plates. After 24 h of cell attachment, the cells were incubated with 20 µg/mL human IgG4 isotype (Biolegend, San Diego, CA, USA), 0.001, 0.01, 0.05, 0.1, 0.25, 0.5, 1, 2.5, 5, 10, and 20 µg/mL nivolumab (OPDIVO^®®^; Bristol-Myers Squibb, US), and 5 µg/mL doxorubicin hydrochloride (D1515; Sigma-Aldrich) for 24, 48, and 72 h. Nivolumab was a generous gift from Dr. Chia-Jui, Yen [Department of Internal Medicine, NCKU Hospital]. After the indicated incubation time, 0.5 mg/mL of 3-(4,5-dimethylthiazol-2-yl)-2,5-diphenyltetrazolium bromide (M5655; Sigma-Aldrich) was added, and the cells were further incubated for 3 h. The formazan crystals were solubilized with dimethyl sulfoxide, and the absorbance was measured using the microplate reader (SpectraMAX 340PC384; Molecular Devices) at a wavelength of 570 nm. The percentage of cell viability was calculated as [(OD_sample_ − OD_blank_/OD_control_ − OD_blank_)] × 100%.

### 2.5. RUES2-CMs Treatment and Stimulation

RUES2-CMs were treated with 20 µg/mL human IgG4 isotype, 0, 0.5, 0.75, 1, 5, 10, 20 µg/mL nivolumab, and 5 µg/mL doxorubicin for 72 h, or, RUES2-CMs was stimulated with 20 ng/mL Recombinant Human IFN-γ (#300-02; PeproTech Inc., Rocky Hill, NJ, USA) for 48 h. The cells were then harvested for evaluating protein expression using Western blot and flow cytometry.

### 2.6. Isolation, Activation, and Expansion of CD4^+^ and CD8^+^ T-lymphocytes

Healthy human PBMC (peripheral blood mononuclear cell) comes from commercial products approved by the Institutional Review Board (70025; Stemcell technology). Thawed PBMCs were cultured in RPMI 1640 supplemented with 10% (fetal bovine serum) FBS and 1% penicillin-streptomycin (pen-strep; GeneDireX, Taipei, Taiwan) in a humidified atmosphere containing 5% CO_2_ at 37 °C.

CD4^+^ and CD8^+^ T-lymphocytes were isolated using CD4^+^ and CD8^+^ negative selection kits (#480010 and #480012; Biolegend) according to manufacturer’s instruction. Briefly, CD4^+^ or CD8^+^ Biotin-Antibody Cocktail was added to PBMCs and incubated for 15 min on ice. Next, CD4^+^ or CD8^+^ streptavidin nanobeads were added, followed by 15-min incubation on ice prior to being exposed to a magnet for 5 min. The liquid with the desired cells was collected. The desired cells (CD4^+^ or CD8^+^ T-lymphocytes) were resuspended in RPMI 1640 supplemented with 10% FBS and 1% pen-strep. T-lymphocytes were activated using human T-activator CD3/CD28 kit (#11161D; Invitrogen) per manufacturer’s instruction. For 1 × 10^6^ T-lymphocytes, 25 µL of Dynabeads^®®^ coated with anti-CD3 and anti-CD28 antibodies was added to the medium supplemented with 100 ng/mL human recombinant IL-2 (#200-02; PeproTech Inc.). Both T-lymphocytes were allowed to expand for 7 days.

### 2.7. Co-culture RUES2-CMs with Isolated CD4^+^ or CD8^+^ T-lymphocytes

A total number of 1.5 × 10^5^ RUES2-CMs were plated on 24-well plates pre-coated with 50 µg/mL Matrigel in RPMI/B27 medium, or 7.5 × 10^5^ activated CD4^+^ or CD8^+^ T-lymphocytes were incubated with 10 ng/mL human IgG, 1, 5, and 10 µg/mL nivolumab, respectively, for 20 min prior to co-culture with RUES2-CMs. Finally, activated CD4+ or CD8+ T-lymphocytes were co-cultured with a 5:1 effector to target ratio with RUES2-CMs and incubated for 72 h prior to analysis.

### 2.8. Enzyme-linked Immunosorbent Assay (ELISA)

The conditioned medium from the co-culture model was collected and centrifuged at 300× *g* to collect the supernatant. IFN-γ cytokines in the harvested supernatant were measured using a commercial ELISA kit in accordance with the manufacturer’s instruction (LEGEND MAX Human IFN-gamma ELISA kit; Biolegend, San Diego, CA, USA).

### 2.9. Flow Cytometry

RUES2 cells were detached using Accutase, fixed, and permeabilized by BD Cytofix/Cytoperm Fixation/Permeabilization solution kit (#554714; BD Biosciences, San Diego, CA, USA). The cells were blocked with 5% BSA and incubated on ice for 30 min with primary antibodies OCT-4

-Alexa Fluor 647, SSEA-4-PE, and Nanog-PE, respectively. Trypsinized RUES2-CMs were washed, fixed, blocked, and permeabilized prior to being incubated with cTnT-Brilliant Violet 421 (BV421) antibody for 30 min on ice. In the co-culture model, RUES2-CMs and T-lymphocytes were separated by washing with PBS prior to the staining process. RUES2-CMs were stained with Annexin V-PE antibody prior to the fixation process and stained with PD-L1-Brilliant Blue 515 (BB515) without permeabilization. Both isolated and activated CD4^+^ and CD8^+^ T-lymphocytes were stained with CD25-APC, PD-1-APC antibodies without permeabilization. All antibodies were purchased from BD Biosciences. All stained samples’ data were acquired on the BD FACSCanto II flow cytometer (BD Biosciences). The data were then analyzed with FlowJo version 10 software (Tree Star; Ashland, OR, USA).

### 2.10. Western Blot

RUES2-CMs’ total protein was extracted using 1×radio immunoprecipitation assay (RIPA) lysis buffer (Millipore, Billerica, MA, USA) supplemented with protease and phosphatase inhibitors (Roche Diagnostics, Mannheim, Germany). The cell lysate at 72 h post-co-culture with immune cells was collected for the analysis of apoptosis, and the sample at 30 min post-co-culture was collected for the detection of a phosphorylated protein. In the co-culture model, RUES2-CMs were separated from T-lymphocytes by washing off T-lymphocytes with PBS prior to protein extraction. Protein concentration was quantified using Bicinchoninic Acid (BCA) Protein Assay kit (G Biosciences, Maryland Heights, MO, USA). A total of 20 µg of proteins was run on 12% SDS-polyacrylamide gel (Bio-Rad, Hercules, CA, USA) and transferred to immobilon-P nitrocellulose membranes (Millipore). The membrane was blocked with 5% skim milk and immunoblotted with primary antibody overnight at 4 °C with gentle agitation. The membranes were probed with the following primary antibodies: phospho-STAT1 (#9177, Cell Signaling Technology, Danvers, MA, USA), STAT1 (9175#, Cell Signaling Technology, Danvers, MA, USA), phospho-NFκB (#3033, Cell Signaling Technology, Danvers, MA, USA), NFκB (#8242, Cell Signaling Technology, Danvers, MA, USA), caspase-3 (#9662, Cell Signaling Technology, Danvers, MA, USA), cleaved-caspase-3 (#9661, Cell Signaling Technology, Danvers, MA, USA), PD-L1 (#17952, Proteintech Group Inc.), and anti-GAPDH (#2118, Cell Signaling Technology, Danvers, MA, USA). The membranes were incubated with appropriate secondary antibodies (goat anti-rabbit and goat anti-mouse IgG-HRP conjugated antibodies; 1:5000; Jackson ImmunoResearch Laboratories, West Grove, PA, USA). The enhanced chemiluminescence substrate (Millipore, Burlington, MA, USA) was used to detect proteins. The membranes were visualized using the iBright FL1000 Imager (Thermo Fisher Scientific, Waltham, MA, USA).

### 2.11. Animals

In this project, the experiments were performed using BALB/cByJNarl mice (*n* = 28, 8-week-old) purchased from the National Laboratory Animal Center, Yilan, Taiwan. All animals were housed in the Laboratory Animal Center, National Cheng Kung University, Tainan, Taiwan, in controlled conditions with a normal 12 h light/dark schedule. All animals were free to access to water and food. All animals were acclimatized for 1 week in an animal laboratory center before starting experiments. All procedures and protocols performed on the animals were approved by the Institutional Animal Care and Use Committee, National Cheng Kung University, Tainan, Taiwan (approved IACUC number: 107212: 20.4.2018-21.4.2021).

### 2.12. B16-F10 Mouse Melanoma Tumor Xenograft Model

All animals were randomly categorized into four groups: (1) non-tumor-bearing IgG treatment group (*n* = 7), (2) non-tumor-bearing anti-PD-1 treatment group (*n* = 7), (3) tumor-bearing IgG treatment group (*n* = 7), and (4) tumor-bearing anti-PD-1 treatment group (*n* = 7). Briefly, 5 × 10^5^ B16-F10 mouse melanoma cells were suspended in 50% Matrigel diluted with DMEM/FBS (#354234; Corning) and implanted subcutaneously into the right dorsal flank of mice in the tumor-bearing group. The non-tumor-bearing group was injected with 50% Matrigel in DMEM/FBS without cells. When tumor volumes reached approximately 100 mm^3^, the mice were injected intraperitoneally with 250 μg rat IgG or anti-PD-1 (RMP1-14, BioXcell, West Lebanon, NH, USA) in respective groups. Both treatments were given every 72 h for a total of six doses. Tumor size was measured twice a week with a digital caliper and calculated using this equation: Tumor volume (mm^3^) = 0.5 × length (longest tumor diameter) × width^2^ (shortest tumor diameter^2^).

### 2.13. Echocardiography

The cardiac function was accessed by echocardiography before treatment and on the sacrifice day. Briefly, mice were anesthetized using isoflurane and placed in the supine position, and the echocardiography was performed using a Vevo 770 (Visual Sonic, Toronto, ON, Canada) imaging system. All examinations were performed by the same blind examiner. The echocardiographic images were obtained from the M-mode image, at the midpapillary muscle level in a parasternal view. The measurements obtained were left ventricular internal diameter at end-diastole (LVID’d), left ventricular internal diameter at end-systole (LVID’s), fraction shortening (FS), and ejection fraction (EF). All of these measurements were analyzed with VEVO analysis software. The ejection fraction percentage was calculated as [(LVEDV − LVESV)/LVEDV] × 100%, while fractional shortening was calculated as [(LVID;d − LVID;s)/Lividly] × 100%.

### 2.14. Mouse Facial Vein Blood Collection

The vascular bundle at the ventral jaw region was set as a landmark for the facial vein. The facial vein of an unanesthetized mouse was punctured using a 5 mm lancet (Mediapoint Inc., Mineola, NY, USA). Approximately 100 μL blood was collected in microcentrifuge tubes pre-coated with 50 mM EDTA. The collected blood was centrifuged at 1500× *g* for 15 min at 4 °C. The plasma supernatant was then carefully transferred to a microcentrifuge tube. The collected plasma samples were stored at −80°C immediately after collection until cTnI analysis by ELISA assay.

### 2.15. Tissue Processing and Immunohistochemistry Staining

Mouse hearts were perfused at the time of scarification and fixed in 4% PFA for 24 h before the dehydration process. The dehydration service was provided by Human Biobank, Research Center of Clinical Medicine, NCKU. The dehydrated tissues were then paraffin-embedded and sliced into 5 μm sections using a microtome (Leica Rotary Microtome RM2235; Leica Biosystems, Buffalo Grove, IL). The sectioned tissues were incubated for 15 min at 65 °C and then immersed in xylene for 5 min twice to remove paraffin. Samples were rehydrated by immersing in 100%, 95%, 80%, and 50% ethanol, each for 3 min. Samples were then stained with hematoxylin for 5 min and eosin for 1 min for morphology observation. Samples were rehydrated and then stained for target proteins with antigen retrieval by soaking the slides in sodium citrate buffer (10 mM, pH 6) for 20 min at 95 °C. After the samples could cool to room temperature, they were immersed in 3% H_2_O_2_ for 10 min at room temperature. After washing thrice with PBST, the samples were blocked with 5% BSA in PBST for 1 h at room temperature. After the blocking process, the samples were then stained with primary antibody overnight at 4 °C with anti-CD8 and anti-CD4 (1:100; Abcam) and anti-PD-L1 (1:100; Invitrogen) antibodies. On the next day, the samples were washed 5x and then a secondary antibody with conjugated HRP (goat anti-rabbit, goat anti-mouse, and goat anti-rat HRP conjugated antibodies, 1:500; Invitrogen) was added to the samples and incubated for 1 h at room temperature. 3,3′Diaminobenzidine (DAB) substrate (Dako, Agilent Technologies, Inc, Glostrup, Denmark) was added as a chromogen to visualize the desired target. Nuclei were counterstained by hematoxylin.

### 2.16. RNA Isolation and Reverse Transcription

Heart tissues were cut into small pieces and collected in microcentrifuge tubes. The 1 mL of TRIzol (Invitrogen) was added to the tubes and homogenized on ice. This was followed by the addition of 200 μL of chloroform to emulsify the sample. The aqueous layer, which contained RNA, was isolated by centrifugation at 14,000 rpm for 20 min at 4 °C. The layer was carefully aspirated and transferred to a new RNase-free microcentrifuge tube. Approximately 800 μL isopropanol was added to each tube and gently mixed. RNA pellets could be observed at the bottom of the tubes after centrifugation at 14,000 rpm for 10 min. Then, 1 mL of 75% ethanol was added to the pellet tubes, gently mixed, and centrifuged to discard the supernatant. This process was repeated twice, and the pellets were allowed to air-dry.

The purity and concentration of the isolated RNA were determined using a NanoDrop 2000 spectrophotometer (Thermo Fisher Scientific, Waltham, MA, USA) by reading the optical density at 260 and 280 nm. The 1 μg of RNA was used to synthesize cDNA for heart tissues. Briefly, 1 μL of 1 nM oligomers, 10 μL of 2 X reverse transcription premix (Yeastern Biotech, Taipei, Taiwan), and 5.28 μL of 1 μg template RNA were added into a PCR tube and heated to 65 °C for 5 min in a PCR machine. Next, 1 μL reverse transcriptase enzyme premix, 0.5 μL RNase inhibitor (RNaseOUT, Thermo Fisher Scientific), and 2.22 μL DEPC-treated (Thermo Fisher Scientific, Waltham, MA, USA) water were added to the mixture for a final volume of 20 μL. Finally, the mixture was incubated at 42 °C for 1 h and 70°C for 15 min. After the reaction was done, the synthesized cDNA was stored immediately at −20 °C.

### 2.17. Polymerase Chain Reaction (PCR) And Real-Time Quantitative PCR (RT-qPCR)

PCR mixture was prepared by adding 10 μL of PCR master mix (Thermo Fisher Scientific), 1 μL 10 μM forward primer, 1 μL 10 mM reverse primer, 1 μL synthesized cDNA, and 7 μL of nuclease-free water. PCR conditions were set as, initial denaturation at 95 °C for 1 min, subsequent denaturation at 95 °C for 30 s, annealing at 60 °C for 30 s, and extension at 72 °C for 1 min, for 35 cycles, and 72 °C final extension for 10 min. RT-PCR was performed in a reaction volume of 10 μL containing 5 μL Fast SYBR green master mix (Applied Biosystem, Foster City, CA, USA), 1 μL of synthesized cDNA, 1 μL 3 nM of each forward and reverse primers, and 2 μL of nuclease-free water. RT-qPCR reaction was performed using the ABI StepOne Real-Time PCR System (Applied Biosystem, Foster City, CA, USA). Relative fold change in gene expression was calculated using the 2(−ΔΔCt) formula. PCR primer sequences for human and mouse PD-L1 were 5′- TTGCTGAACGCCCCATACAA-3′ (forward) and 5′-TCAGTGCTACACCAAGGCAT-3′(reverse) and 5′-AAGTCAATGCCCCATACCGC-3′ (forward) and 5′-TTCTGGATAACCCTCGGCCT-3′ (reverse). RT-PCR primer sequences for mouse IFN-γ and TNF-α were 5′-CAGCAACAGCAAGGCGAAAA-3′ (forward) and 5′-TCATTGAATGCTTGGCGCTG-3′ (reverse) and 5′-ACTGAACTTCGGGGTGATCG-3′ (forward) and 5′-CCACTTGGTGGTTTGTGAGTG-3′ (reverse).

### 2.18. Statistical Analysis

All data are expressed as the mean ± standard deviation (SD). The effects of nivolumab on RUES2-CM in the MTT assay, flow cytometry, quantification of western blot, and cytokine studies were analyzed by one-way analysis of variance (ANOVA), followed by Bonferroni’s posthoc tests using GraphPad Prism 6 software (GraphPad Software, Inc, San Diego, CA, USA). The RUES2-CMs IFN-γ stimulation experiments were analyzed using unpaired two-tailed Student’s t-test. The statistical significance was considered at *p*-value < 0.05.

## 3. Results

### 3.1. Generation and Characterization of hESC-derived Cardiomyocytes

Cardiomyocytes were generated from a human embryonic stem cell line, RUES2, using the established cardiomyocyte protocol of GSK3 inhibitor (chir99021) and Wnt signaling inhibitor (XAV939) treatment, then incubating the cells in the base medium of RPMI containing B27 without insulin until day 7. At this time, the medium was changed to use RPMI containing B27 with an insulin supplement. Approximately 8 days after induction, spontaneously contracting cells were observed. On day 9, 2% B-27 with insulin was added to glucose-free RPMI 1640 medium for the selection of cardiomyocytes. After 4 days of selection, the cells were harvested and disassociated into single cardiomyocytes with trypsin for further experiments (Figure 1A). Day 14 RUES2-derived cardiomyocytes were used for the following experiments: Flow cytometry analysis revealed that over 91% of RUES2 cells exhibited triple-positive pluripotency markers Oct-4, SSEA-4, and Nanog expression but did not express cardiac troponin T (cTnT), a marker of differentiated cardiomyocytes in mammals. Day 14 RUES2-derived cardiomyocytes (RUES2-CM) differentiated to express high levels of cTnT, and the pluripotency markers were down-regulated (Figure 1B). Immunostaining demonstrated that the RUES2 cells expressed the pluripotency markers Oct-4, and the nuclear marker DAPI was used to identify nuclear localization (top panel; Figure 1C), and RUES2-CM cells were double positive for the cardiac markers α-actinin and cTnT (down panel; Figure 1C).

To investigate the role of PD-1 in regulating cardiac inflammation and damage, we first analyzed the expression of PD-L1 on RUES2-CMs in vitro. Figure 1D shows that flow cytometric analysis revealed the expression of PD-L1 in RUES2-CMs. Additionally, RUES2-CMs double-stained for cardiomyocyte markers cTnT (red) and PD-L1 (green) (Figure 1E).

### 3.2. Nivolumab Treatment Did Not Induce Apoptosis and Inflammation in Cardiomyocytes

Next, we investigated the effects of the nivolumab treatment on cell viability in RUES2-CMs. We observed that exposure of RUES2-CMs to nivolumab for 72 h did not significantly affect cell viability in the MTT assay (Figure 2A). Doxorubicin-treated RUES2-CMs for 72 h were used as a positive control. Doxorubicin is a well-known cardiotoxic chemotherapy agent [15], and the condition confirmed to induce cell death (Figure 2A). We then assessed cell apoptosis using the TUNEL assay and flow cytometric analysis (Annexin V, early apoptosis marker) and compared to cells treated with doxorubicin for an equal time as a positive control. No apoptotic cell was detected after 72 h in nivolumab-treated RUES2-CMs (Figure 2B,C). Therefore, in subsequent experiments, doxorubicin was used as a positive control.

Western blot showed the activation of caspase-3 in cell lysates from doxorubicin-treated RUES2-CMs but not in nivolumab-treated RUES2-CMs (top panel; Figure 2D,E). Furthermore, phospho-STAT1 and phospho-NFκB did not show statistically significant up-regulation in response to nivolumab treatment in RUES2-CMs (down panel; Figure 2D,E).

Next, we determined whether nivolumab treatment induced an increase in PD-L1 expression and inflammatory signal protein expression in RUES2-CMs. Flow cytometric analysis showed no statistically significant up-regulation of PD-L1 in response to nivolumab treatment in RUES2-CMs compared to doxorubicin-treated RUES2-CMs (Figure 2F).

### 3.3. Increase of PD-1 Expression on CD4^+^ T-lymphocytes and CD8^+^ T-lymphocytes upon Activation

To examine the involvement of T-lymphocytes in the PD-1/PD-L1 pathway-induced autoimmune myocarditis, PD-1 was expressed in CD4^+^ (upper/left panel, Figure 3A) and CD8^+^ (left/lower panel, Figure 3A) T-lymphocytes. To determine whether PD-1 expression was up-regulated upon T-lymphocytes activation, expression of T-cell activation marker CD25 was analyzed by flow cytometry. Expression of CD25 was upregulated in CD4^+^ (upper/right panel, Figure 3A) and CD8^+^ (lower/right panel, Figure 3A) T-lymphocytes after anti-CD3/CD28 antibody stimulation. Following the activation of T-lymphocytes, PD-1 expression was up-regulated in activated CD4^+^ (upper panel, Figure 3B) and CD8^+^ (lower panel, Figure 3B) T-lymphocytes as detected by immunofluorescence staining.

To examine the effects of nivolumab treatment on cardiac PD-L1, flow cytometry was used in subsequent experiments. The expression of PD-L1 from RUES2-CMs was significantly increased after the nivolumab treatment of CD4^+^ T-lymphocyte and RUES2-CM in the co-culture model (Figure 3C). By contrast, the addition of the nivolumab at different doses had no discernible effect on PD-L1 production in CD8^+^ T-lymphocytes and RUES2-CMs co-culture model. The inflammation-related protein expression in CD4^+^/CD8^+^ T-lymphocytes and RUES2-CMs co-culture model was also assessed. Significantly elevated RUES2-CMs production levels of inflammation-related proteins were found in CD4^+^ T-lymphocytes and RUES2-CMs co-culture model but not in CD8^+^ T-lymphocytes and RUES2-CMs co-culture model (Figure 3D–G).

There is evidence that the interaction of PD-1 with PD-L1 down-regulates T-lymphocyte cytokine production [16]. To test whether the effects of nivolumab treatment on stimulation of pro-inflammatory cytokine was cell-contact dependent, we co-cultured CD4^+^ and CD8^+^ T-lymphocytes with RUES2-CMs at an effector to target cell ratio of 5:1 for 72 h and assessed IFN-γ production by ELISA. Co-culturing CD4^+^ and CD8^+^ T-lymphocytes with RUES2-CMs resulted in a significant reduction in IFN-γ production by the T-lymphocytes (Figure 3H). To test whether the PD-L1-derived inhibition of IFN-γ production was due to PD-L1:PD-1 interaction, nivolumab or human IgG4 isotype control was added to the co-culture systems. The addition of nivolumab significantly restored the IFN-γ production by the CD4^+^ T-lymphocytes but not CD8^+^ T-lymphocytes (Figure 3H).

### 3.4. Anti-PD-1 Immunotherapy Increases Myocardium T Lymphocytes Infiltration in Tumor-Bearing Mice

Due to a majority of reported cardiotoxic cases occurring in melanoma patients receiving anti-PD-1 ICI, the B16-F10 melanoma cell line was utilized to investigate the efficacy of anti-PD-1 antibodies in treating cancer. A tumor-bearing mice model was established by inoculating B16-F10 mouse melanoma cells subcutaneously to examine the anti-tumor and cardiotoxic events of anti-PD-1 treatment. We first evaluated the PD-L1 expression of the B16-F10 cell line and the mouse heart to confirm the validity of the model prior to giving anti-PD-1 antibody treatment (Figure 4A–E). Next, we confirmed PD-L1 was expressed by B16-F10 melanoma cells (Figure 4A) and the mouse heart using immunostaining for protein (Figure 4B–E) and reverse transcriptase PCR for mRNA (Figure 4F).

As shown in Figure 4G, we subcutaneously inoculated B16-F10 murine melanoma cells into the flank of male BALB/c mice. When the size of the tumors reached approximately 100 mm^3^ (~9 days), the mice were randomly assigned to either a rat IgG control or an anti-PD-1 treatment group. The tumors were excised after a 27-day growth period. Anti-PD-1 treatment significantly decreased tumor size, whereas the treatment with rat IgG did not affect the tumor growth (Figure 4H). 

Parameters of left ventricle systolic function, for example, fractional shortening and ejection fraction, were measured. As shown in Figure 4I,J, the FS and EF were significantly decreased in the anti-PD-1 treated-tumor group compared to IgG-treated tumor group (FS = 22.62% versus 28.42%, (*p* < 0.05); EF = 44.58% versus 53.36%, (*p* < 0.05)). There were no statistically significant differences in heart function parameters between the rat IgG-treated non-tumor group and anti-PD-1-treated non-tumor group (Figure 4I,J). After echocardiography, mice were euthanized instead of histological analyses. Infiltrating immune cells in the cardiac tissue were examined by immunofluorescence analyses after anti-PD-1-treated tumor-bearing mice.

The infiltration of extensive CD4^+^ (Figure 4K; upper panel) and CD8^+^ (Figure 4K; lower panel) T lymphocytes was detected in the anti-PD-1-treated tumor group (Figure 4K). Western blot showed the anti-PD-1 immunotherapy significantly increased CD4 and CD8 expression in the anti-PD-1-treated tumor group as compared with the IgG-treated tumor group (Figure 4L,M). However, there were no significant differences between the IgG-treated non-tumor group and anti-PD-1-treated non-tumor group (Figure 4L,M).

### 3.5. Anti-PD-1 Immunotherapy Induces Cardiac Inflammation and Apoptosis in Tumor-Bearing Mice

To determine whether the reduced heart function was secondary to injury of cardiomyocytes and the subsequent cardiac inflammation and damage, we extracted the protein from the whole heart tissue of mice to perform western blot analysis of protein expression. As shown in Figure 5A,B, significantly increased inflammation-related proteins were found in the anti-PD-1-treated tumor group compared to the IgG-treated tumor group. In addition, mRNA expression of IFN-γ and TNF-α in the heart tissue increased significantly in the anti-PD-1-treated tumor group compared to the anti-PD-1-treated non-tumor group (Figure 5C). Significantly decreased p-Akt and increased cleaved-caspase-3 were found in the anti-PD-1-treated tumor group compared to the rat IgG-treated tumor group (Figure 5D,E). 

## 4. Discussion

We provided an in vitro platform to study cardiomyopathy mediated by nivolumab using hESC-CMs, RUES2-CMs. hESC-CMs have become excellent in vitro platform for evaluating drug toxicity on cardiomyocytes owing to its robust generation and relevant physiological phenotype of human cardiomyocytes [17]. In addition, nivolumab is a fully-humanized anti-human-PD-1 antibody; therefore, RUES2-CMs serve as an appropriate in vitro model in evaluating nivolumab-induced cardiomyopathy.

We first validated PD-L1 expression in RUES2-CMs prior to subsequent toxicity screening experiments involving the PD-1/PD-L1 interaction. Because there is no in vitro evidence to show a direct effect of nivolumab on cardiomyocytes, we tested the toxicity of nivolumab alone in RUES2-CMs. The results showed no toxicity on RUES2-CM even when incubated with a high dosage of nivolumab. This indicated that cardiomyopathy mediated by nivolumab required an additional component.

To further understand the mechanism of nivolumab on cardiomyocytes, we chose to involve T-lymphocytes into the in vitro model. Nivolumab targets mainly PD-1 on T-lymphocytes to prevent its ligation with PD-L1 on tumor cells, and hereafter prevent tumor cells from escaping immune surveillance. In addition, previous in vitro studies had utilized human T-lymphocytes, co-cultured with tumor cells to evaluate the efficacy of checkpoint blockade in treating cancer [18,19], thereby we included T-lymphocytes in our in vitro model. This idea was also supported by myocardium T-lymphocytes infiltration in PD-1^-/-^ and PD-L1^-/-^ knockout mice [13,20]. Furthermore, clinical immune-mediated myocarditis displayed massive T-lymphocyte infiltration to the myocardium after receiving ICIs treatment [6,21]. Here, we chose the direct contact co-culture model instead of a transwell co-culture model because PD-1 on T-lymphocytes requires direct contact with PD-L1 on RUES2-CMs. Furthermore, in our previous pilot study of a RUES2-CMs/Jurkat cell co-culture model, nivolumab increased pro-inflammatory cytokines in the direct contact model but not in the transwell model (data not shown).

Cytokine level is an important criterion to represent T-cells activity; thus, we chose to test the pro-inflammatory cytokine, IFN-γ level, to evaluate T-cell function under the influence of nivolumab.

Furthermore, our results displayed that IFN-γ treatment was able to increase RUES2-CMs PD-L1 expression (data not shown). From the cytokine results, the co-culture of activated CD4^+^ and CD8^+^ T-lymphocytes with RUES2-CMs resulted in significantly reduced IFN-γ production when compared to activated CD4^+^ and CD8^+^ T-lymphocytes alone. This might be due to increased expression of PD-1 level on activated CD4^+^ and CD8^+^ T-lymphocytes; this increased PD-1 ligation with PD-L1 on cardiomyocytes and delivered immune inhibitory signals. To test whether the reduction of IFN-γ was modulated by the PD-1/PD-L1 axis, we included nivolumab in the co-culture model. Nivolumab targeted PD-1 to prevent its ligation with PD-L1, and thus inhibitory signals were removed, and T-lymphocytes were activated to elicit an immune response that acted against cardiomyocytes. This was supported by the recovery of cytokine production when nivolumab was added to CD4^+^ T-lymphocytes co-culture model. Hence, the results supported that the PD-1/PD-L1 axis played a role in down-regulating T-cell response against cardiomyocytes.

To test whether RUES2-CMs underwent inflammation, two inflammatory mediators—NFκB and STAT1—were chosen because NFκB and STAT1 are able to regulate PD-L1 expression in cancer cells [22]. An increase in NFκB signaling has been shown to enhance myocardial inflammation [23], and inhibiting this pathway alleviates virus-induced myocarditis [24]. On the other hand, STAT1, downstream signaling of IFN-γ, exacerbates mouse myocarditis [25]. Thus, our results showed that the expression of phosphorylated NFκB and STAT1 was increased along with IFN-γ level after treatment with nivolumab in the CD4^+^ T-lymphocytes co-culture model.

The expression of PD-L1 on cardiomyocytes is up-regulated when cardiomyocytes are injured [26], and our results showed that the PD-L1 level was increased when treated with doxorubicin (Figure 2F). Additionally, cleaved caspase-3, an apoptosis marker, was increased concurrently with PD-L1 level in RUES2-CMs after CD4^+^ T-lymphocytes co-cultured with nivolumab treatment. These results indicated that PD-L1 expression was correlated with cardiomyocyte injuries, such as inflammation and apoptosis. It is believed this up-regulation serves as a protective mechanism but is abrogated by inhibition of immune checkpoint [6].

Interestingly, CD4^+^ T-lymphocytes dominated the inflammatory and apoptotic responses, but CD8^+^ T-lymphocytes played a lesser role in inducing nivolumab-mediated cardiomyopathy. CD8^+^ T-lymphocytes did not respond to nivolumab treatment, showing any significant increase in IFN-γ cytokine levels, inflammatory mediators, and no induction of apoptosis after treatment. In the CD8^+^ T-lymphocytes co-culture model, phosphorylated STAT1 and NFκB were mildly increased after adding nivolumab, demonstrating CD8^+^ T-lymphocytes still played an effector role in cardiomyocyte inflammation and apoptosis. Previous evidence has also supported that CD8^+^ T-lymphocytes mediate PD-1/PD-L1-deficient or inhibition-related myocarditis [27,28].

The possible reason that CD4^+^ T-lymphocytes dominated over CD8^+^ T-lymphocytes in cardiac inflammatory and apoptotic response is the CD4^+^ T-lymphocytes’ response during the initiation and early stages of myocarditis. In autoimmune myocarditis progression, CD4^+^ T-lymphocytes serve as the main driving force. This is due to CD4^+^ T-lymphocytes being induced by IFN-γ to the CD4^+^ T-helper type 1 (Th1) subset, and subsequently, CD4^+^ Th1 cells are required to promote the activation of autoimmune CD8^+^ effector cell-mediated cardiac damage [29]. The depletion of CD4^+^ T-lymphocytes has shown reduced cytotoxicity in myocarditis and CD8^+^ T-lymphocytes activities [30]. Furthermore, CD4^+^ T-lymphocytes are believed to be able to generate effector responses in the absence of CD8^+^ T-lymphocytes, but the mechanisms are still unclear. Studies have shown that CD4^+^ T-lymphocytes are able to induce tissue graph cytopathic effects in acute allograft rejection without involving CD8^+^ T-lymphocytes. The proposed mechanisms of the effector response of CD4^+^ T-lymphocytes include the pro-inflammatory cytokines—TNF-α and IFN-γ secretion—and activate the innate immune response [31]. In our co-culture model, CD8^+^ T-lymphocytes produced less IFN-γ cytokine compared to CD4^+^ T-lymphocytes, and this might be the reason for lacking an inflammatory and apoptotic effector response by CD8^+^ T-lymphocytes. Increasing CD8^+^ T-lymphocytes effector cell ratio, prolonging nivolumab incubation time, or co-culturing RUES2-CMs with both CD4^+^ T-lymphocytes and CD8^+^ T-lymphocytes together might elicit CD8^+^ effector T-lymphocytes-mediated cardiac damage activities and exacerbate cardiomyocytes’ inflammation and apoptosis. 

## 5. Conclusions

Our results demonstrated a potential CD4^+^ T-lymphocytes-mediated inflammation and apoptosis in cardiomyocytes via blocking PD-1/PD-L1. We indicated that PD-L1 expression on cardiomyocytes played an important role in peripheral immune tolerance. Inhibition of PD-1/PD-L1 ligation on T-lymphocytes by nivolumab led to T-lymphocyte activation and subsequently gave rise to T-lymphocyte-mediated autoimmune cardiomyopathy.

### 5.1. Study Limitations

The limitation of the present study is that the in vitro nivolumab treatment does not mimic the administration route taken by cancer patients. The dosage and treatment period is not compatible with clinical implementation. With respect to the in vivo study, although the risk of cardiotoxic effects of anti-PD-1 antibody by evaluating cardiac contractile function and histology presented in the heart of mice, the physiological characteristics, heart rate, drug metabolism rate, and immune cell composition cannot fully represent the human condition. We expect to address these issues with more experimentation and by more clinical evidence.

Future studies may include T-lymphocytes of patients who underwent immune-related myocarditis in co-culture models to better understand the difference of T-cell function during cardiomyopathy when compared to the healthy donor.

### 5.2. Ethical Approval and Consent to Participate

This is not a clinical trial design. All procedures and protocols performed on animals were approved by Institutional Animal Care and Use Committee, National Cheng Kung University, Tainan, Taiwan (approved IACUC number: 107212). PBMCs were purchased from a licensed company with an IRB approval. Healthy human PBMC comes from commercial products approved by the Institutional Review Board (70025; Stemcell technology).

## Figures and Tables

**Figure 1 ijms-21-02399-f001:**
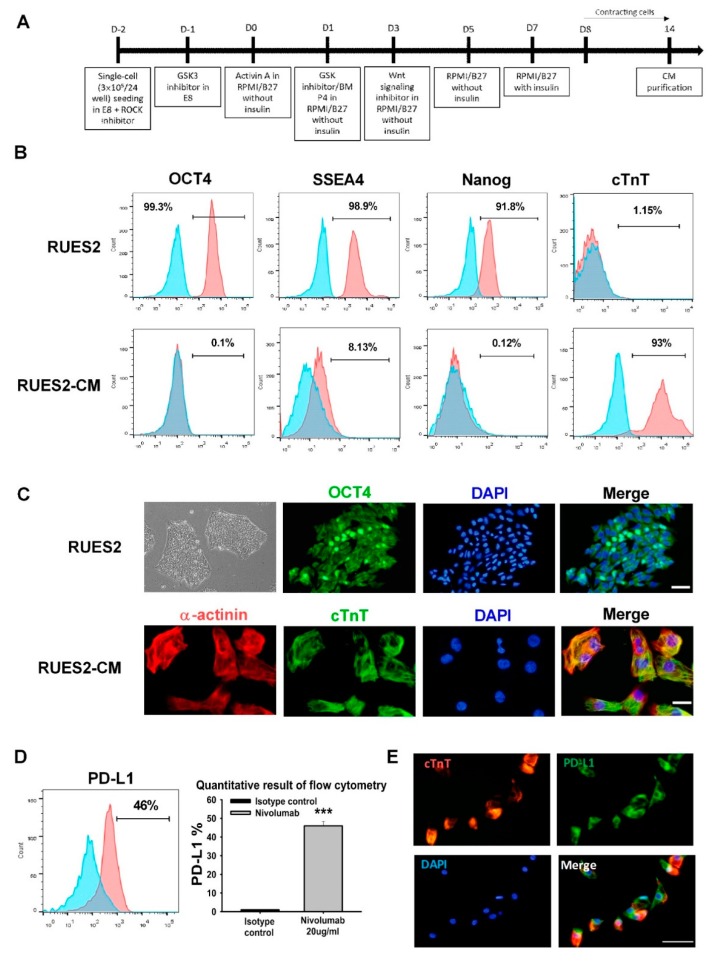
Establishment and validation of RUES2 and differentiation of cardiomyocytes in vitro. (**A**) Schematic of the RUES2-CMs differentiation protocol. (**B**) Flow cytometric analysis of pluripotent stem cell markers expression: Oct-4, SSEA-4, Nanog, and a positively selected cardiac marker (cTnT). (**C**) Representative fluorescent images of pluripotent stem cell markers Oct-4 and cardiac markers: α-actinin and cTnT. Scale bar: 50 μm (upper panel); 20 μm (lower panel). (**D**) PD-L1 expression on RUES2-CMs by flow cytometry and (**E**) immunofluorescence staining, scale bar: 25 μm. The quantitative result as the right panel. *** *p* < 0.001 versus control (*n* = 3), one-way ANOVA, posthoc Bonferroni test. RUES2, Rockefeller University embryonic stem cell line 2; cTnT, cardiac troponin T; Oct4, octamer-binding transcription factor 4; PD-L1, programmed death-ligand 1; RUES2-CMs, Rockefeller University embryonic stem cell line 2-cardiomyocytes.

**Figure 2 ijms-21-02399-f002:**
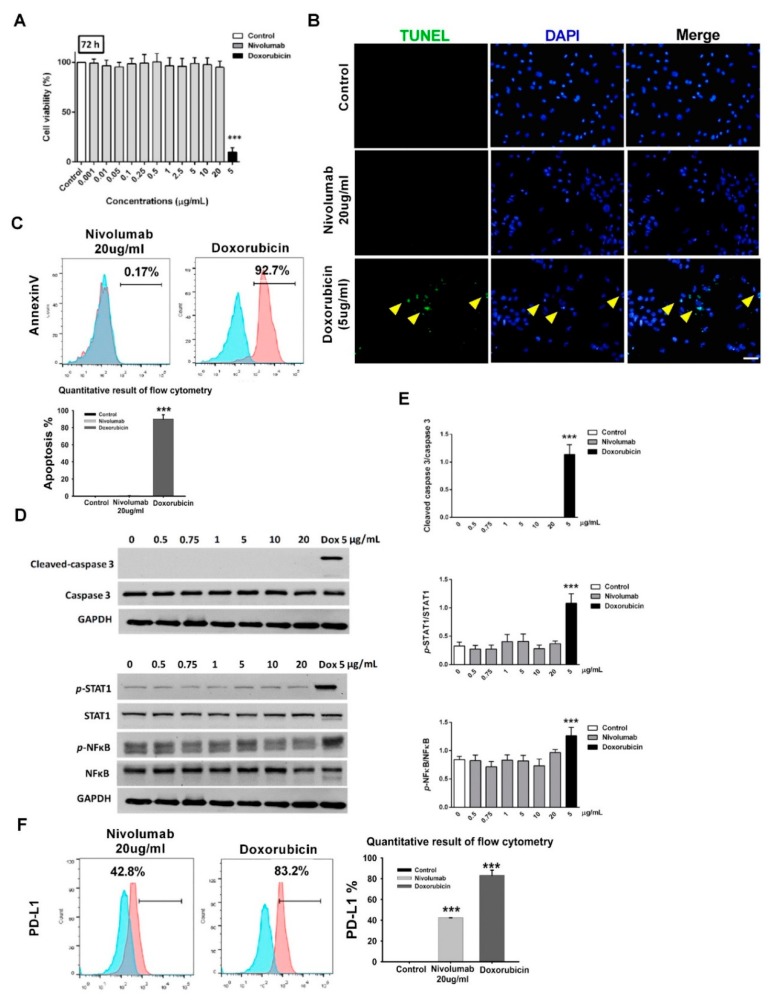
Effects of nivolumab on cell viability in RUES2-CMs. (**A**) The effect of nivolumab on cell viability was determined by an MTT assay. Positive control (doxorubicin 5 μg/mL) was provided. *** *p* < 0.001 versus control (*n* = 3). (**B**) Representative images of the TUNEL staining of cardiomyocytes. TUNEL staining was used to detect cell apoptosis (green). The nuclei were counterstained with DAPI (blue). Cyan color represents TUNEL-positive nuclei on merged photos. Positive control (doxorubicin 5 μg/mL) was provided. Scale bar: 50 μm. (**C**) Representative flow cytometry images of the Annexin V. Positive control (doxorubicin 5 μg/mL) was provided. *** *p* < 0.001 versus control (*n* = 3), one-way ANOVA, posthoc Bonferroni test. (**D**) Representative western blot analysis of caspase3 and inflammation markers—STAT1, NFkB—and the quantitative result (**E**). Positive control (doxorubicin 5 μg/mL) was provided. *** *p* < 0.001 versus control (*n* = 3), one-way ANOVA, posthoc Bonferroni test. Data are shown as the mean ± SD. RUES2-CM, Rockefeller University embryonic stem cell line 2- cardiomyocytes. (**F**) The PD-L1 expression level in RUES2-CMs analyzed by flow cytometry images. The quantitative result is shown in the right panel. *** *p* < 0.001 versus control (*n* = 3), one-way ANOVA, posthoc Bonferroni test.

**Figure 3 ijms-21-02399-f003:**
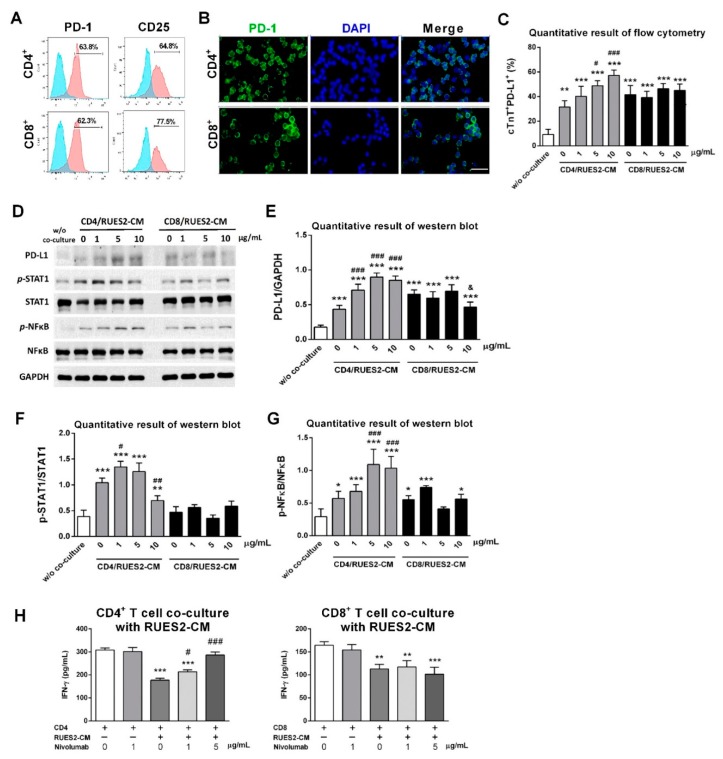
Validation of PD-1 expression on CD4^+^ and CD8^+^ T lymphocytes. (**A**) Representative flow cytometry results of PD-1 and activation marker CD25 expression on CD4+ and CD8+ T lymphocytes after anti-CD3/CD28 stimulation. PD-1, programmed death-1. (**B**) Representative micrographs showing immunostained PD-1^+^ cells (green) on the CD4^+^ and CD8^+^ T lymphocytes. Scale bar: 50 μm. (**C**) Quantification of cTnT^+^/PD-L1 expression by flow cytometry analysis of levels of cTnT^+^/PD-L1 expression on RUES2-CMs co-cultured with CD4^+^ and CD8^+^ T lymphocytes with nivolumab treatment at different doses. ** *p* < 0.01, *** *p* < 0.001 versus RUES2-CMs alone group; # *p* < 0.05, ### *p* < 0.001 versus CD4/RUES2-CMs human IgG4 isotype control (w/o co-culture) group (*n* = 3). (**D**) Representative western blot analysis on RUES2-CMs co-cultured with CD4^+^ and CD8^+^ T lymphocytes containing nivolumab treatment with different doses and quantitative results (**E**–**G**). * *p* < 0.05, ** *p* < 0.01, *** *p* < 0.001 versus RUES2-CMs alone group; # *p* < 0.05, ## *p* < 0.01, ### *p* < 0.001 versus CD4/RUES2-CMs human IgG4 isotype control (w/o co-culture) group (n=3; & *p* < 0.05 versus CD8/RUES2-CMs human IgG4 isotype control (w/o co-culture) group; & *p* < 0.05, versus CD8/RUES2-CMs human IgG4 isotype control (w/o co-culture) group (*n* = 3), one-way ANOVA, posthoc Bonferroni test. (**H**) Quantitative result of ELISA assay, showing production of IFN-γ in CD4^+^ T lymphocytes/RUES2-CMs and CD8^+^ T lymphocytes/RUES2-CMs co-culture medium. ** *p* < 0.01, *** *p* < 0.001 versus CD4+ T lymphocytes alone group; # *p* < 0.05, ### *p* < 0.001 versus CD4/RUES2-CMs human IgG4 isotype control (w/o co-culture) group (*n* = 3).

**Figure 4 ijms-21-02399-f004:**
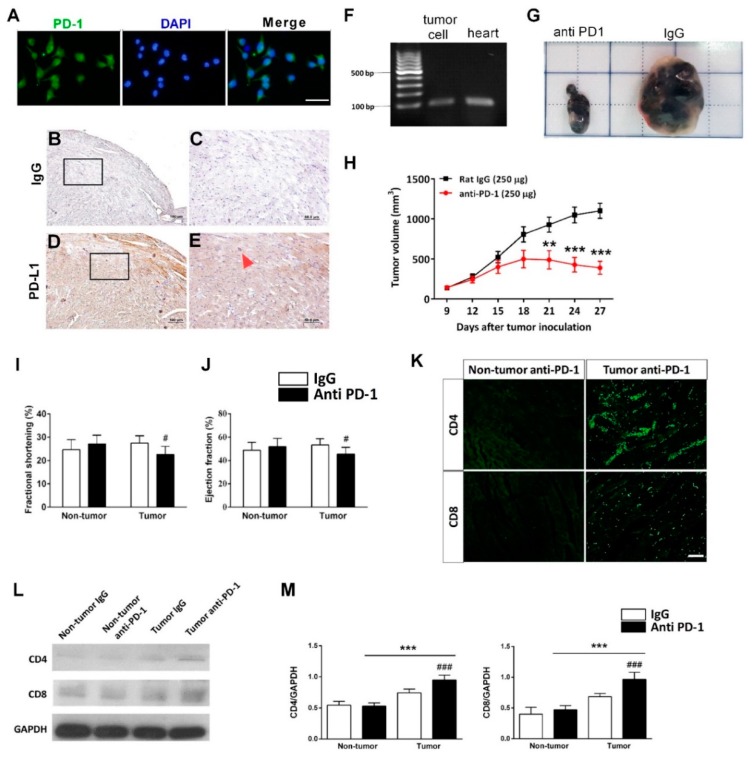
Effects of anti-PD-1 immunotherapy on tumor growth in tumor-bearing mice. (**A**) Representative micrographs showing immunostained PD-L1^+^ cells (green) on the B16-F10 cells. Scale bar: 50 μm. Representative immunohistochemistry staining for IgG (**B**,**C**) and PD-L1 (**D**,**E**) on mouse heart. The inset box, showing the enlarged pictures for the indicated square area. Scale bars in (B) and (D), 100 μm; scale bars in (C) and (E), 50 μm. Red arrow: PD-L1^+^ cell. (**F**) Representative PCR analysis of PD-L1 in B16-F10 cells and mouse heart. PD-L1, programmed death-ligand 1. (**G**) Representative photograph of xenograft tumors at the end of the experiment. (**H**) Tumor growth curves for each experiment group. ** *p* < 0.01, *** *p* < 0.001 versus rat IgG group (*n* = 7), two-way ANOVA, posthoc Bonferroni test. Effects of anti-PD-1 immunotherapy on mouse heart function. The quantitative data of fractional shortening (**I**), and ejection fraction (**J**) of rat IgG and anti-PD-1-treated group. # *p* < 0.05 versus rat IgG-treated tumor group (*n* = 7); # *p* < 0.05 versus anti-PD-1-treated non-tumor group (*n* = 7). (**K**) Representative micrographs showing immunostained CD4^+^ and CD8^+^ cells (green) in the myocardium. Scale bar: 1000 μm. (**L**) Representative western blot analysis and quantitative results (**M**). *** *p* < 0.001 versus anti-PD-1-treated non-tumor group; ### *p* < 0.001 versus rat IgG-treated tumor group (*n* = 4), two-way ANOVA, posthoc Bonferroni test. Data are shown as the mean ± SD.

**Figure 5 ijms-21-02399-f005:**
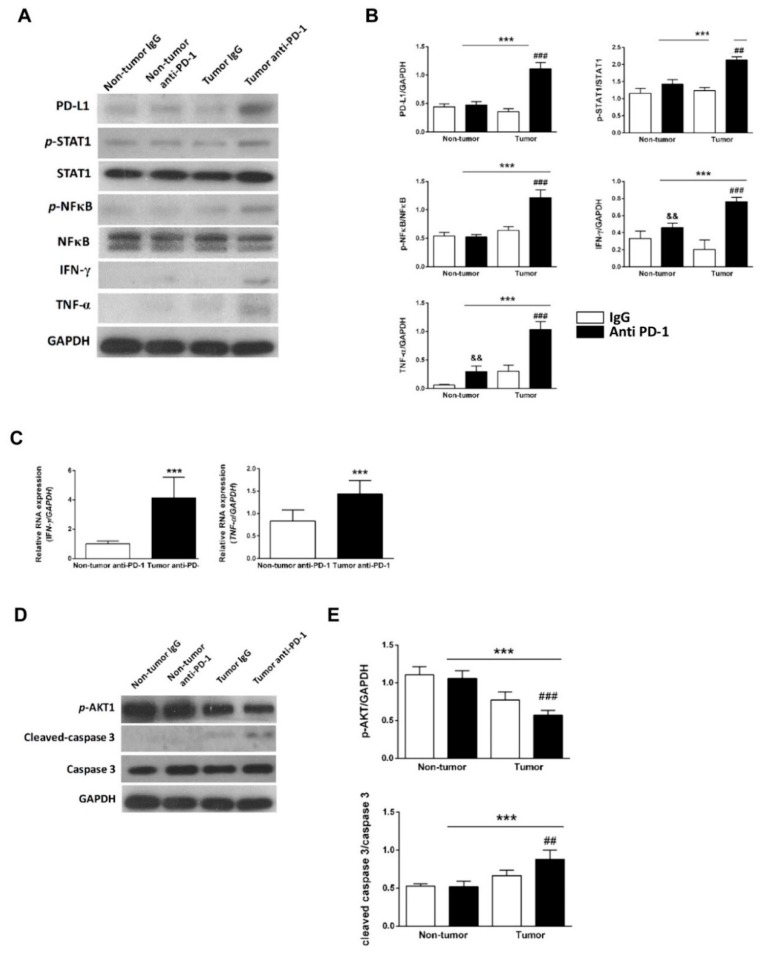
Effects of anti-PD-1 immunotherapy on the expression of PD-L1 and inflammation-related markers in tumor-bearing mice. (**A**) Representative western blot analysis and quantitative results (**B**). && *p* < 0.01, ### *p* < 0.001 versus rat IgG-treated tumor group; *** *p* < 0.001 versus anti-PD-1-treated non-tumor group; && *p* < 0.01 versus rat IgG-treated non-tumor group (*n* = 4), two-way ANOVA, posthoc Bonferroni test. (**C**) A representative of RT-PCR analysis of IFN-γ and TNF-α in the anti-PD-1-treated non-tumor group and anti-PD-1-treated tumor group. *** *p* < 0.001 versus anti-PD-1-treated non-tumor group (*n* = 4), unpaired Student’s t-test. Data are shown as the mean ± SD. (**D**) Representative western blot analysis of apoptosis markers and quantitative results (**E**). ## *p* < 0.01, ### *p* < 0.001 versus rat IgG-treated tumor group; *** *p* < 0.001 versus anti-PD-1-treated non-tumor group (*n* = 4), two-way ANOVA, posthoc Bonferroni test.

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
