# Peer review of "Programmed Cell Death-1: Programmed Cell Death-Ligand 1 Interaction Protects Human Cardiomyocytes Against T-Cell Mediated Inflammation and Apoptosis Response In Vitro"

_ijms, 2020, doi:10.3390/ijms21072399_

Round 1

Reviewer 1 Report

In this study, Tay and coworkers analysed cardiotoxic effects of nivolumab, which is a monoclonal antibody used in cancer therapies, on cardiomyocytes. For this purpose, authors used RUES2 cells that were differentiated into cardiomyocytes, in combination with CD4 or CD8 positive T-lymphocytes in co-culture experiments.

First of all, authors characterize efficient differentiation of RUES2 cells into cardiomyocytes (Fig. 1).

The characterization results are convincing. However, the presentation of the data is difficult and confusing. The terms used in the method section should be the same as in the result section. But this is not the case. In the results it is said “using well established protocol by treat of GSK3 inhibitor and Wnt signaling inhibitor”. However, in the method section this is not mentioned, or are they the same as CHIR99021 or XAV939??? Please clarify.

Furthermore, time points of analysis are not given in Fig. 1 or the result section. Please add!

In Fig. 2 effects of nivolumab in comparison to doxorubicin on cardiomyocyte viability are analysed. While nivolumab does not cause cell death or immunological responses, the positive control does. Also here authors should indicate the time points. MTT and annexin assays were done after 72 h. Was the same time point used for western blots?

Fig. 2A needs a figure legend, similar to 2E, indicating control, nivolumab, and doxorubicin columns.

Fig. 3 analyses immunresponses in T-lymphocytes upon activation. Activation of lymphocytes was achieved by anti CD3/CD28 stimulation. This results in enhanced expression of PD-1 and CD25. In co-cultures of CMs and lymphocytes CD4 positive cells PD-L1, p-STAT1 and p-NFkB expression increase in presence of nivolumab. A decrease in INF-gamma release is found in CD4-CMs co-cultures that is reversed by nivolumab. These results bear some problems.

  1. What cells were used? In the methods a commercial source of cells is mentioned, in the results section isolation of PBMCs from blood is described.
  2. What is fig. 3C presenting? In the legends PDL-1 expression is said, in the y-axis cTnT PDL1 is given.
  3. In fig. 3 C-G use of nivolumab must be stated.
  4. Please define the control group precisely. Does w/o co-culture mean in presence of control IgG4?
  5. On page 9, lines 282-292: What is meant with “Central Illustration”???

What is the difference between co-cultures in Fig. 3 C-G vs. Fig. 3 H?  For Fig. H use of target to cell ratio was 5:1. Was this different to the other parts of Fig. 3? Could authors present a picture of the co-cluture to ensure that cell-cell contacts were present, as they state.

Fig. 3 I is mentioned in the result section, but does not exist in reality.

In Fig. 4 effects of anti-PD-1 therapy in tumor-bearing mice is presented. Reduction of tumor progression is demonstrated under anti-PD-1 therapy, as well as reduced cardiac function in echo analysis is presented, and enhanced lymphocyte infiltration in the heart is shown.

Fig. 4G is described in the result section as showing subcutaneous inoculation of melanoma cells into mice. However, Fig. 4G presents tumor size in controls and under therapy. It is unclear, at what time point tumors were excised.

Fig 4F: In the legends statistics are indicated. However, this is just a representative PCR gel without need for statistics.

Fig. 5 presents enhanced apoptosis and pro-inflammatory responses in tumor bearing mice under therapy.

I suspect that the analyses are performed on heart tissue. However, it is not stated anywhere.

While the data are convincing, authors themselves completely forget their own in vivo data in the abstract and in the discussion. In the discussion, they even add in the limitations of the study, that their findings are completely done in vitro, and heart function could not be assessed. But in Fig. 4 fractional shortening and EF is shown. In the ethical approval they state that no animals were applied in their study and nothing about the animals is said in the method section.

The data presentation is in some parts not straightforward and difficult to follow. Many spelling and grammar errors make reading even more difficult.

Reviewer 2 Report

The article by Tay et al. entitled ,, Programmed cell death-1: programmed cell death- 2 ligand 1 interaction protects human cardiomyocytes against T-cell mediated inflammation and apoptosis response in vitro’’ demonstrates the effect of nivolumab treatment on RUES2-CM cells and investigates the potential cardiotoxic effect of T cell response of PD-1/PD-L1 axis in  regulating cardiomyocytes injury in vitro. Overall, I think that Authors made an effort to conduct this research, however, I think that the manuscript is not written in a clear way, so I have some concerns regarding this work. My detailed comments are given below:

  1. Since the nature of these studies is not so straightforward a schematic representation of the main assumptions/findings in a form of Graphical Abstract or a Figure in the text, would be highly appreciated.
  2. Abstract section: please reconstruct this section, since ,,Translational Perspective’’ part is out of place here.
  3. Materials and Methods: lack of description of animal-based experiments? Figure 4 contains data obtained from mice-based research, and no description is provided in the Materials and Methods section. Same with PCR analysis, Fig. 4 and 5 show PCR-based results and PCR is not even mentioned in the M&M section.
  4. Results: please present all the FACS results on bar graphs with indication of statistical significance. Current way of presentation is not clear enough, FACS images may stay however, as graphical representation of cytometric results.
  5. Results: line 283: what does the ,,Central Illustration’’ refer to here?
  6. Results: Fig.3 panel B – please provide the descriptions of the photos CD4+/CD8+, panel C – why cTnT+PD-L1+ is indicated on the Y axis of the graph and only PD-L1 is referenced in the figure legend?, panel H – current presentation of this figure would suggest external addition of the CD4 and CD8 to the culture, which I do not think is the case here, please remove this and maybe put cell names above the graphs.
  7. Results: line 331: only panels are referenced, but no indication of which Figure?
  8. Results: Fig. 4 B – please provide arrows indicating stains of interest, panel G – would be nice to see also starting size of tumors, not only the end result, to get fuller insight into the results of immunotherapy with anti-PD1.
  9. Discussion: line 374: in vivo platform?
  10. Discussion: line 400: where are these results shown?
  11. English grammar and spelling need to be thoroughly corrected, since the number of mistakes is countless, e.g. line 23: ,, Abstract: Aim:  Immnological  checkpoint (…)’’ – Immunological, line 32: ,, Cell viability were confirmed after treatment.’’ – was, line 32-33: ,,The immune response were evalute by using CD4+ or CD8+ T-lymphocytes, first to activate them in vitro, then co-cultured with RUES2-CMs in the presence of nivolumab for.’’ – the whole sentence is just wrong, etc. Please use some help from the native spiker or professional English correcting service.
  12. I think the Conclusions are very diluted and unclear. These are basically results in brief. Stronger emphasis of researchers’ idea of these studies should be put in this section. Maybe translational perspective should be placed here to somehow summarize the results and give the idea of their utility from the pharmacological/oncopharmacological/cardiological perspective?
  13. In the Introduction/Discussion, some emphasis should be put into the possible practical use of the investigated phenomenon e.g. what are the key points of the results, and how they can be translated into clinical practice in the future?
  14. Also, a thorough proofread of the manuscript is necessary since there are some formatting mistakes and typos, e.g. Figure 4 panel K, there is a letter ,,M’’ next to the CD8 signature, also Figure 2 panel A, the graph has the ,,Concentration’’ description placed twice, but no sign of what treatment was applied – should be given here to make the graph self-explanatory (also first mention of bar captions are far away on panel E)
  15. I am also a little bit confused with the ,,Ethical Approval…’’ part, stating that ,,(…) no animals were applied during the experiments.’’ Where there are clearly results from mice experiments. Please explain.

Round 2

Reviewer 1 Report

Authors fixed the problems of the first version. Most important, the animal experiments are now adequately described and perfectly complement the in vitro data.

Reviewer 2 Report

I think the manuscript has been significantly corrected.